# Food Volume Estimation Based on Deep Learning View Synthesis from a Single Depth Map

**DOI:** 10.3390/nu10122005

**Published:** 2018-12-18

**Authors:** Frank P. -W. Lo, Yingnan Sun, Jianing Qiu, Benny Lo

**Affiliations:** 1Hamlyn Centre, Department of Surgery and Cancer, Imperial College London, London SW7 2AZ, UK; benny.lo@imperial.ac.uk; 2Hamlyn Centre, Department of Computing, Imperial College London, London SW7 2AZ, UK; y.sun16@imperial.ac.uk (Y.S.); jianing.qiu17@imperial.ac.uk (J.Q.)

**Keywords:** dietary assessment, volume estimation, mhealth, deep learning, view synthesis, image rendering, 3d reconstruction

## Abstract

An objective dietary assessment system can help users to understand their dietary behavior and enable targeted interventions to address underlying health problems. To accurately quantify dietary intake, measurement of the portion size or food volume is required. For volume estimation, previous research studies mostly focused on using model-based or stereo-based approaches which rely on manual intervention or require users to capture multiple frames from different viewing angles which can be tedious. In this paper, a view synthesis approach based on deep learning is proposed to reconstruct 3D point clouds of food items and estimate the volume from a single depth image. A distinct neural network is designed to use a depth image from one viewing angle to predict another depth image captured from the corresponding opposite viewing angle. The whole 3D point cloud map is then reconstructed by fusing the initial data points with the synthesized points of the object items through the proposed point cloud completion and Iterative Closest Point (ICP) algorithms. Furthermore, a database with depth images of food object items captured from different viewing angles is constructed with image rendering and used to validate the proposed neural network. The methodology is then evaluated by comparing the volume estimated by the synthesized 3D point cloud with the ground truth volume of the object items.

## 1. Introduction

In nutritional epidemiology, detailed food information is required to help dietitians to evaluate the eating behavior of participants. To measure food intake, 24-hour dietary recall (24HR), a dietary assessment method, is commonly used to capture information on the food eaten by the participants. A complete dietary assessment procedure can mainly be divided into four parts: food identification, portion size estimation, nutrient intake calculations, and dietary analysis. When using 24HR, however, the volume or the portion size of object items relies heavily on participants’ subjective judgement which undoubtedly leads to inaccurate and biased dietary assessment results. Thus, it is essential to develop objective dietary assessment techniques to address the problems of inaccuracy and subjective measurements. Due to the advances in computer vision, dietary assessment technology has significantly been improved recently, paving the way for accurate dietary analysis [1,2]. For volume estimation, a variety of computer vision-based techniques have also been proposed to tackle the problem of quantifying food portions. The food volume measurement techniques can be divided into two main categories: model-based and stereo-based techniques. For instance, Sun et al. [3] proposed a virtual reality (VR) approach which utilizes pre-built 3D food models with known volumes for users to superimpose onto the food items in the real scene. By scaling, translating, and rotating the models to fit the food items in the image, the food volumes can then be estimated. This proposed technique has a root mean square error of around 20.5%. A similar idea was also presented in [4] where pre-built models from a model library were used to match and register the object items. With such modelling techniques, the error rate ranges from 3.6 to 12.3%. Several previous studies by the research group from Purdue University [5,6,7] have also been published. Zhu et al. proposed two models, a spherical and a prismatic approximation model, for estimating portion size. For spherical food items, the spherical model can be used to estimate volume through projecting the feature points (elliptical region) onto the plane surface. For the food items with irregular shapes, an assumption that the masked area of the object items should correspond to the contact area of the plane surface can be made, and the prismatic approximation model can then be used to estimate the volume. The proposed method has an error rate ranging from 2.03 to 10.45%. For the stereo-based approach, Puri et al. [8] proposed a food volume estimation technique based on a multi-view reconstruction method. By capturing images from different viewing angles, extrinsic camera calibration can then be carried out to determine the relative camera positions. In order to perform extrinsic calibration, feature matching can be performed with RANdom SAmple Consensus (RANSAC). The algorithm has been applied and evaluated on several food objects and the error in volume estimation ranges from 2.0 to 9.5%. One of the biggest hurdles presented with this approach is its slow processing time—it takes approximately 33 seconds to carry out a single 3D reconstruction. Recently, Dehais et al. [9] presented the idea of using a two-view 3D reconstruction to further increase the processing speed. A modified RANSAC algorithm was proposed to carry out relative pose extraction. When using this approach, the percentage error ranges from 8.2 to 9.8%. It was evaluated on general food object items and outperformed other stereo-based methodologies. Despite the great performance shown in previous studies, these techniques still have some challenging unsolved problems which include the following: (1) The model-based techniques usually involve different levels of human intervention which require participants to rotate, shift, and scale the pre-built food models to fit them with the food items in the images. (2) The stereo-based approach requires participants to capture multiple food images from different viewing angles which could be tedious. (3) Other methods require feature point extraction and matching. For food objects with smooth surfaces or less significant texture, feature points cannot be extracted effectively which leads to failure in 3D reconstruction. (4) The reflective light of an object can vary when the images are captured from different viewing angles which hinders the accuracy of feature point matching and 3D reconstruction. (5) Reference objects such as fiducial markers are often required to be placed next to the food items for accurate estimation.

With regard to these issues, several novel dietary assessment techniques have been proposed with the advances in artificial intelligence in recent years. Regarding the coupled nature of the volume and depth of object items, a research team from Google designed a framework for using the deep learning approach to estimate the volume of food objects. The underlying principle is to use only a single RGB image to estimate the volume by reconstructing 3D food models based on the inferred depth maps trained by convolutional neural networks (CNN) [10]. The proposed idea can potentially improve the efficiency of assessing dietary intake without requiring participants to capture multiple images from various viewing angles or to place the fiducial marker next to the object items. Despite the convenience of accessing dietary intake with a single RGB image, a wide range of hurdles and challenges sill exist in this field which are described as follows: (1) The performance of using RGB images to infer depth maps is not yet satisfactory and their proposed method always induces a large volume estimation error. (2) The contours of the object items in the inferred depth map are blurred which largely affects the volume estimation as well. Although much research has been conducted on depth estimation based on deep learning, these experiments have mostly been carried out to evaluate the performance of the neural network in perceiving relative distances or the 3D relationships between object items from the environment. Most of these approaches can provide an approximation to the distance only in the presence of rich environmental cues, i.e., vanishing points. This results in large volume estimation error when applied in dietary assessment, especially in the situation where the images are captured from close range or bird’s eye view. Regarding the poor performance of RGB images in inferring depth maps, we hypothesize that another potential underlying reason is most likely due to the scale ambiguity of a single RGB image which makes the neural network difficult to converge. On the other hand, occlusion in volume estimation by using a single image is also a challenging issue. It was shown in [11] that occlusion often leads to overestimation in volume, since the global point cloud or voxel representation reconstructed by bird’s eye view depth image can solely be used to handle food objects with a narrow top and wide bottom. Thus, it is also essential to achieve point cloud completion to restore the structure of the occluded area before carrying out volume measurement of the object items. For more detailed information about different approaches, an overall comparison among state-of-the-art food volume estimation techniques is shown in Table 1.

With the advances in camera technologies, various 3D sensing research teams or companies, including Apple PrimeSense, Intel RealSense, Google Project Tango, and Microsoft’s Kinect [14], have focused on developing depth-sensing cameras that are tiny enough to be embedded in mobile devices. With sensors coming from PrimeSense, Apple launched a sophisticated Face ID unlocking system using human facial characteristics based on depth maps [15]. The advent of depth-sensing has literally changed every aspect of the way people use their smart devices and has enabled the introduction of new depth-enabled applications. Hence, a depth camera-based technique could be the potential solution to quantify dietary intake in a more efficient and accurate way in the coming future. In this paper, an integrated dietary assessment technique based on depth sensing and deep learning view synthesis is proposed. The aim of this technique is to enable food volume estimation by taking a single depth image from any convenient viewing angle. This method is mainly targeted at estimating the volumes of food object items (1) without significant external characteristics or features (2) with occlusion (3) with no fiducial marker needed (4) and could be of irregular shape. To estimate the volume, a new neural network architecture is designed and trained with depth images from the initial image and its corresponding opposite viewing angles (i.e., the front and back views). The whole 3D point cloud map can be reconstructed by fusing the initial data points with the synthesized points of the occluded areas of the object items, and back-projected to the same world coordinate using the inferred camera extrinsic matrix. A modified Iterative Closest Point (ICP) algorithm is also designed to minimize the distance between the initial and synthesized clouds of points so as to further improve the accuracy in food volume estimation. Once the point cloud map is completed, meshing is carried out to estimate the volume of the object items. To examine the feasibility and accuracy of the proposed architecture, a new dataset is constructed by image rendering. Depth images of object items captured from initial image and its corresponding opposite viewing angles are rendered for training and testing.

To the best of our knowledge, this paper is the first study to use deep learning view synthesis to tackle the problem of occluded view and to perform volume estimation. Since there is no prior work, this paper aims to conduct a preliminary study to evaluate the feasibility, efficiency, and accuracy of using deep learning to estimate object volume under the circumstance of occluded views. Thus, most of the experiments were designed for the research investigation and are limited to laboratory settings. Nevertheless, to facilitate the development of objective dietary assessment techniques to enhance the performance of dietary assessment, a pilot study would be an important step forward. The main contributions of this paper can be summarized as follows: (1) The effectiveness of a depth camera-based dietary assessment technique is examined. (2) A comparison among different start-of-the-art food volume estimation approaches is summarized (3) A novel neural network framework is proposed to reconstruct 3D point clouds of different food object items based on view synthesis. (4) A new database is constructed through image rendering to evaluate the performance of the network architecture. (5) A modified ICP algorithm is proposed to enhance the performance of 3D reconstruction. The rest of the paper is organized as follows: Section 2 details the methodology of image rendering, the neural network architecture, and our proposed volume estimation techniques. Section 3 presents the experimental results of the proposed point cloud completion and ICP algorithms in volume estimation. The discussion is presented in Section 4. The paper concludes in Section 5.

## 2. Detailed Information and Methods

To access dietary intake, a variety of aforementioned visual-based dietary assessment methods have been proposed which can be further classified into two categories: image-assisted and image-based approaches. Compared to image-assisted approaches, image-based approaches can be more objective and unbiased in assessing users’ nutritional intake as they do not require much human intervention, such as manual image recognition and analysis. In this paper, a novel image-based technique using deep learning and depth sensing technique is proposed to address several long-standing problems in the field of image-based dietary assessment in which view occlusion and scale ambiguity are major challenges. In this section, the detailed information and methods about deep learning view synthesis are discussed as follows: (1) The procedure of using deep learning view synthesis to perform dietary assessment is presented to show how the proposed technique works. (2) Image rendering is followed to demonstrate how the new database is constructed to evaluate the proposed architecture in volume estimation. (3) The proposed network architecture for depth image prediction is explained. (4) A modified point cloud completion algorithm is then presented to show how the initial and inferred depth images are registered and reconstructed into a complete global point cloud. (5) A new ICP algorithm is shown to optimize the performance of point cloud registration and to enhance the accuracy in volume estimation. (6) Three-dimensional meshing, a well known technique for volume measurement, is discussed.

### 2.1. The Procedure of Deep Learning View Synthesis Approach

The procedure of using deep learning view synthesis to access dietary intake is presented in the following text, and the system diagram is shown in Figure 1: (1) A mobile phone with depth sensors or a depth camera (Intel RealSense/Microsoft Kinect) is used. (2) RGB and depth images are captured from any convenient viewing angle. Note that RGB and depth images should be synchronized. (3) RGB image should be segmented and classified based on the image segmentation approach (ex. Mask R-CNN [16]) (4) The corresponding regions in the depth image are labeled accordingly. (5) Point cloud completion with the deep learning view synthesis is then applied to each labeled object item to perform 3D reconstruction and estimate food volume as shown in Figure 2. (6) Once the volume is measured, the data information can be linked to USDA national nutrient database for further dietary analysis [17]. Note that this work mainly focuses on the application of deep learning view synthesis in volume estimation, so food classification and dietary analysis are not discussed in this paper.

### 2.2. Image Rendering

In order to assess the feasibility of using deep learning view synthesis to quantify dietary intake, a new dataset (https://www.doc.ic.ac.uk/~ys4315/Food_dataset.zip) was constructed through image rendering based on virtual 3D models of the real-life objects from the Yale–CMU–Berkeley object set [18]. A single depth image of the object item is far from enough to reconstruct a complete global point cloud map due to the limited viewing angle. To complete the point cloud of the object, it is essential to obtain prior knowledge of the general object shape. In other words, we need to train a neural network that is able to understand what the back of an object will look like even there is only a single viewing angle. To address this issue, the typical approach is to use a vast amount of virtual 3D models to train the network through 3D CNN and represent the object items using a voxel volume representation. However, this approach is computationally intensive in training and limits the resolution of the inferred 3D models of the object items (usually less than 32 × 32 × 32 voxels) [19]. In this case, the performance of volume estimation will be largely affected since the contours of the object items are blurred/rough due to low resolution. Inspired by [19], this paper proposes a deep learning view synthesis approach to address the issue. Depth images, known as 2.5D images, of object items captured from initial and corresponding opposite viewing angles were rendered to be a training dataset. With the inferred depth images, a completed 3D point cloud of the targeted object item was obtained through registering the camera coordinates of different depth images into a same world coordinate. Various virtual 3D models of real-life objects with 64 k vertex points from the Yale–CMU–Berkeley object set were chosen to be the training model. Each object item was then placed at the origin, and depth images were captured from different viewing angles with 4 major types of movement, azimuthal rotation, elevation rotation, height adjustment, and center shifting, as shown in Figure 3. In training, the depth images from initial viewing angle were randomly rendered with a range of extrinsic camera parameters as shown in Table 2. For each food item, 20,000 pairs of depth images (initial and its corresponding opposite viewing angles) were rendered from various viewing angles for training.

### 2.3. Neural Network Architecture

In order to accurately quantify dietary food intake from images, it is important to address the issue of occlusion. With the proposed approach, the issue of tackling self-occluded food object items can be regarded as a view synthesis problem. With the advances of artificial intelligence, several research groups have proposed view synthesis based on the deep neural network [20,21,22,23]. However, to the best of our knowledge, there has still been no research conducted on object volume estimation based on view synthesis. With a 3D database of sufficient size, a neural network can implicitly learn the 3D shapes of different object items and further explore how the objects will look like with different viewing angles. Besides, with the generalization capability of deep learning, the network can predict the results of using input images from an unseen viewing angle or even using unseen object items. In this approach, the occluded object items can be restored at any convenient viewing angle, and this undoubtedly facilitates accurate quantification of dietary intake.

The proposed network architecture, which was inspired by [20], is shown in Figure 4. The depth image with dimensions 480∗640 is as the input (Intel RealSense). Without going directly into a single convolutional layer, as shown in Figure 4A, the image is processed by 2 inception layers, with 3 convolutional layers of different kernel sizes concatenated together for each of them, as shown in Figure 4B. This specific design is aimed at tackling object items captured from different distances to the object items. Since the sizes of object items may vary, inception layers with kernels of different sizes can capture object details in a easier and more efficient way. After the inception layers, several convolutional layers and fully connected layers are followed to form an image encoder which codes the input image into a vector representation. The underlying reason for this design is to further speed up the training process without having to process the layers in dimensions of 480∗640. Apart from the vector representation of the depth image, the view camera pose, also known as the extrinsic parameter, is also estimated at this stage. This parameter is used to fuse the initial and synthetic depth images so as to complete the occluded parts of the object items in the global point cloud map. On the other hand, the vector representation of the depth image is then directed into several convolutional layers which are known as the image decoder. Finally, the output image is the inferred depth image of dimensions 480∗640. The mathematical expression for the loss function is defined as follows:Cost function for depth image prediction:
(1)$$\sum_{u}^{h}\sum_{v}^{w}{\left(\tilde{d}\left(u,v\right)-d\left(u,v\right)\right)}^{2}+\lambda {\left(\tilde{t}-t\right)}^{2}$$
where $\tilde{d}\left(u,v\right)$ and d(u,v) refer to the pixels from the estimated depth image and the ground truth of depth image, respectively. *w* and *h* refer to the width and height of the image, respectively. λ is the regularization term added to the cost function to better train the neural network. Noted that λ is determined empirically. The selection of λ will affect the convergence speed of the network ($\lambda =10^{-3}$ is used in the cost function). $\tilde{t}$ and *t* refer to the estimated extrinsic parameter (translation matrix) and the ground truth of the extrinsic parameter, respectively.

### 2.4. Point Cloud Completion

Given the inferred depth image, a completed 3D point cloud of the targeted object item can be obtained by registering the camera coordinates of the initial and opposite depth images into the same world coordinate. By doing this, the traditional method is to firstly obtain the extrinsic calibration matrices of the initial image. Once the extrinsic matrices have been determined, the synthetic points can be fused with initial points to obtain a completed point cloud through a transformation matrix. However, the extrinsic matrices are unobtainable without the presence of fiducial markers in normal situations. With regard to this issue, a simple technique is proposed to reconstruct 3D point clouds without the use of fiducial markers. First, the position of the origin should be moved to the center of the initial camera so that the depth image can be reprojected into a world coordinate which is shown in Equation (Equation 2):(2)$$\left[\begin{array}{c} X\\ Y\\ Z \end{array}\right]=ZK^{-1}\left[\begin{array}{c} u\\ v\\ 1 \end{array}\right]\;\;and\;K=\left[\begin{array}{ccc} f_{x} & 0 & c_{x}\\ 0 & f_{y} & c_{y}\\ 0 & 0 & 1 \end{array}\right]$$
where u,v refer to the coordinates in the image and *X*, *Y*, and *Z* refer to the coordinates in the world coordinate. *Z* is a scalar number which refers to depthmap(u,v), and $K\in R^{3x3}$ refers to the intrinsic camera matrix. Second, the position of the opposite camera can be computed easily by carrying out a 180 degree camera rotation and translation through a rotation and a translation matrix, respectively. Given the assumption that the position of the origin has been moved to the center of the initial camera, the angle of the rotation matrix along y-axis can be set to 180 degrees and the matrix can be simplified as Equation (Equation 3):(3)$$R_{y}\left(\theta \right)=\left[\begin{array}{ccc} cos(\theta ) & 0 & sin(\theta )\\ 0 & 1 & 0\\ -sin(\theta ) & 0 & cos(\theta ) \end{array}\right]=\left[\begin{array}{ccc} -1 & 0 & 0\\ 0 & 1 & 0\\ 0 & 0 & -1 \end{array}\right]$$
where θ is the rotation angle of the camera along the y-axis. The translation matrix refers to the translation between the initial and opposite camera positions. This extrinsic parameter can be obtained through the proposed neural network as mentioned in the previous part. Once the rotation and translation matrices have both been obtained, the synthetic point cloud can be registered to the same world coordinates by Equation (Equation 4).
(4)$$\left[\begin{array}{c} X\\ Y\\ Z \end{array}\right]=R^{-1}\left(ZK^{-1}\left[\begin{array}{c} u\\ v\\ 1 \end{array}\right]-T\right)$$
where $R\in R^{3x3}$ refers to the rotation matrix and $T\in R^{3x1}$ represents the translation matrix.

### 2.5. Iterative Closest Point (ICP)

In order to optimize the performance of point cloud registration, a modified ICP algorithm was developed to address the problem of misalignment of initial and synthetic point clouds. From Section 2.3, it is known that the extrinsic parameters obtained through the proposed neural network which will always have a certain estimation error. This error largely affects the volume estimation since the misaligned point clouds will be meshed into a totally different shape. Thus, it is essential to find a solution for better estimating the extrinsic parameters. To address this issue, the estimation problem was regarded as a constrained optimization problem and a modified ICP algorithm was developed to further minimize the difference between the two sets of point clouds so that the translation matrix, which represents the shifting between the center points of two cameras, could be estimated through iterative optimization. Before applying the proposed ICP algorithm, a series of pre-processing techniques are required. (1) Similar to the approach used in [24,25], a bilateral filter is used to fill in the missing spatial information of inferred depth maps. (2) The contours of the object items in both of the input and inferred depth maps are masked out and the corresponding data points are reprojected to the 3D world coordinate as shown in Figure 5. (3) A statistical outlier removal filter is then applied to remove the outliers from the data points. Despite the robust performance of ICP in point cloud registration, it is also known to be susceptible to local minima. Its performance heavily relies on how the temporary transformation matrix is initialized [26]. Taking this issue into account, we propose the use of the inferred extrinsic parameters to initialize the temporary matrix. Given the point clouds and the initialization parameters, ICP can then be used to map the data points through iterations to find the optimized registration solution. Apart from this, the traditional method to perform ICP is to keep estimating the rotation matrix and translation matrix using a root-mean-square point-to-point distance metric minimization technique. On the contrary, in our approach, the formula for iterative optimization is constrained without considering the update of the rotation matrix. This is because the object pose has already been trained and shown in the inferred depth image where the position and depth values of the object item represent the pose configuration. Thus, the relationship between the initial and corresponding opposite depth images can be seen as a simple translation is shown as Equation (Equation 5):(5)$$\begin{array}{c} \left[\begin{array}{c} X_{t+1}\\ Y_{t+1}\\ Z_{t+1} \end{array}\right]=\left[\begin{array}{c} X_{t}\\ Y_{t}\\ Z_{t} \end{array}\right]+T \end{array}$$
where $T\in R^{3x1}$ represents the translation matrix, $X_{t}$, $Y_{t}$, and $Z_{t}$ refer to coordinates in the world coordinates in iteration *t*. In this approach, the optimization procedure is carried out by estimating the translation matrix using a root-mean-square point-to-point distance as a cost function. This cost function is minimized through the use of a gradient descent so that a translation matrix can be determined in which each source point best aligns to its match found in the previous step. Once the transformation has been determined, the synthetic data points can then be transformed and aligned with the initial data points as shown in Figure 5.

### 2.6. Meshing

After the point cloud map has been completed, the next step is to estimate the volume of the food items. The common technique to carry out volume estimation is to mesh the object item based on the convex hull algorithm [12,27,28]. In using this approach, however, object items should be assumed to be convex which largely affects the performance of volume estimation. On the contrary, the alpha shape is applied in our approach which has no limitation on the shapes of object items [29,30]. By using alpha shapes, a sphere with a fixed radius should firstly be defined and a starting point should be chosen from the contours of the object items. The sphere is then rotated with its circumference around the object item from the starting point until the sphere hits another point on the contour. The sphere is then transfered to this point and the process repeats until the loop closes. In Figure 6, the global point cloud of a 3D banana model has been converted into a 3D mesh based on the alpha shape. Once the 3D mesh has been obtained, the volumes of the object items can be easily estimated.

## 3. Experimental Results

### 3.1. Performance of Depth Estimation Based on the Modified Encoder-Decoder Neural Network

In our experiment, 3D models with irregular shapes, including a banana, orange, pear, cube, potted meat can, lemon, tuna fish can and a pudding box, from the Yale–CMU–Berkeley object dataset were chosen to explore the feasibility of our integrated dietary assessment approach. The image rendering technique was used to render depth images for each of the object items. To examine the feasibility of the proposed network in inferring depth image from unseen viewing angles and to avoid over-fitting, it is essential to carry out a thorough evaluation. Similar to the method used in previous works of point cloud completion [20,31,32], the holdout method, a simple kind of cross validation, was used to evaluate the performance of the model in which 70% (20 k images per object item) of the rendered depth images were used to train the neural network, while 10% (2.85 k images per object item) and 20% (5.71 k images per object item) of the images with unseen viewing angles were selected as the validation dataset and testing dataset, respectively. The advantage of this validation method is that it shows the generative ability of the approach without the computationally expensive process of training and re-training large convolution networks. To compare the results of depth estimation with our proposed architecture and the naive version, the training and testing loss are listed in Table 3. From the table, it is shown that the proposed network with inception layers and extrinsic parameter prediction outperformed the naive version and the network without inception layers in both the training and testing sets. Furthermore, the graphs of training and testing loss versus iterations for the naive version and the proposed version (with inception layers and extrinsic parameter) are plotted respectively in Figure 7A,B. Both figures show that the testing loss was comparable to the training loss which indicates that the trained models are generic and able to tackle the images captured from unseen viewing angles without over-fitting. The graphs of testing loss versus iterations for both of the models are plotted together to evaluate the improvement of the network with inception layers and extrinsic parameters as shown in Figure 7C. From the figure, we can see that the loss of the naive model dropped and converged faster than the proposed model. Nevertheless, the testing loss of the proposed model was lower after millions of iterations while the naive model saturated in the middle of the training. From the experimental results, it is possible to conclude that the network with different kernel size and extrinsic parameters prediction can implicitly learn the object details in a more efficient way and which leads to improvement of the accuracy of depth estimation. In addition, the inferred depth images based on the proposed architecture are shown in Figure 8.

### 3.2. Accuracy of Volume Estimation Based on Point Cloud Completion and ICP Algorithms

In the field of view synthesis, most of the experimental evaluations that have been carried out in previous works have only involved comparisons on a qualitative basis. The reason for this is that there is still no conclusion as to which quantitative measure is the most suitable one to evaluate the quality of the generated images. On the contrary, with the use of the depth image as the image input in this study, the scale of the object items can be determined. Thus, the volume can be used to evaluate the performance of the proposed algorithms quantitatively. A comparison of the volume estimation result of our approach with the ground truth to determine the accuracy of the volume estimation method is shown in Table 4. Given the dimensions (height, width, and length) of the object items in the Yale–CMU–Berkeley object dataset, the ground truth volume can be computed based on the given object dimensions. For the objects with a general shape, such as the orange, cube, tuna fish can and pudding box, the volume was easily computed based on geometric calculations. For irregular objects, however, the volume could not be accurately computed using the dimensions provided. It is for this reason that we used the mean estimated volume $V_{g}$ as another reference volume to evaluate the performance of the proposed algorithms for objects without ground truth volume as prior knowledge. The mean estimated volume $V_{g}$ was computed based on the point cloud completion algorithm (listed in Equations (Equation 2)–(Equation 4)) using the known extrinsic parameters generated during image rendering. Fifteen trials were carried out for each object item. As shown in the table, the results of the volume estimation based on point cloud completion were promising with only a 2.4% error (accuracy of 97.6%). This error was obtained by taking the average of the error of the volume estimation-based on point cloud completion for different object items with ground truth volume. This robust performance implies that this technique can be used as a reference to measure the volume of object items without the ground truth. The mean estimated volume $V_{p}$ was computed based on the ICP algorithm (listed in Equation (Equation 5)) using the extrinsic parameters estimated by the proposed neural network and optimized by the ICP algorithm. The error of $V_{p}$ was compared between the ground truth volume and the estimated volume $V_{p}$ when the ground truth volume was provided. For the object items without ground truth, the error was computed using the mean estimated volume $V_{g}$ and mean estimated volume $V_{p}$. The overall performances of the proposed algorithms are shown in Table 4. Small standard deviations in the error can be found in the results due to different initial viewing angles and the graph of estimated volumes with standard deviations is shown in Figure 9. Furthermore, in order to better visualize the performance of the point cloud completion algorithms, the experimental results of 3D reconstruction on chosen object items with unseen captured viewing angles are presented in Figure 10. In addition, the processing time for carrying out point cloud completion was examined in this work to evaluate its real-time operability. The whole procedure took 0.84±0.02 s to estimate the volume of a single food object, thus outperforming stereo-based volume estimation methods.

## 4. Discussion

Several important issues related to image-based food volume estimation have been raised, such as view occlusion, scale ambiguity, and feature extraction problems. These challenging issues have long been discussed in the field of vision-based dietary assessment, yet there are still no solutions that can address all of the mentioned problems. For instance, the stereo-based approach is one commonly used technique to measure food portion sizes. This approach relies strongly on feature matching between frames in which the volumes of certain food items with deformable shapes can be measured. In using the stereo-based approach, a larger variety of food items can be estimated compared to other approaches. However, one major concern of using the stereo-based approach is that the 3D models cannot be reconstructed and the volume estimation will fail if the food surface does not have distinctive characteristics or texture (e.g., fruits, rice). Furthermore, another concern is that it requires users to capture multiple images from certain viewing angles, which, in turn, makes this approach very tedious to use. These findings show that dietary assessment based on a single image seems to be one of the future trends in dietary assessment. As we know, the model-based approach, which is based on a single image, is one of the most reliable volume estimation methods nowadays. Despite the robust performance of this approach, it involves high levels of human intervention and requires participants to rotate, shift, and scale the pre-built food models to find the models that best fit the food items in the images. In addition, another drawback is that the pre-built food models library usually consists of 3D models with simple shapes only, such as sphere, cylinder and hemisphere in which the performance of volume estimation is largely affected when food items are of irregular shapes. In this paper, a deep learning view synthesis approach is proposed to address these major issues in food volume estimation. Through evaluating with food items of irregular geometric shape and non-obvious external features, the experimental results show the promising performance of the proposed technique in volume estimation with accuracy levels of up to 97.6% and 93.1% (using the proposed point cloud completion and ICP algorithms respectively). These findings show that the proposed method outperforms the state-of-the-art volume estimation techniques. This implies that the combination of the depth-sensing technique and deep learning view synthesis has strong potential for accurate image-based dietary assessment. For view occlusion, the proposed algorithms have shown the ability to restore the self-occluded part and provide a complete point cloud of the object items. Nevertheless, one of the problems worth looking into is that we found that the neural network cannot efficiently handle object items with serious center shifting, as shown in Figure 3. The object item in the inferred depth image will appear in an inaccurate position which affects the following point cloud completion. It is for this reason that an inaccurate position of the object item refers to a wrong rotation matrix. In future work, we are planning to train the neural network with more input parameters such as by giving the center points of the items in order to speed up the convergence of the network. For scale ambiguity, we proved the feasibility of using the depth camera to obtain the actual distance information and assess food intake without fiducial markers. This finding shows that depth-sensing technique could have growing potential for accurate dietary assessment. In addition, since this is the first study to use deep learning view synthesis to estimate food volume, most of the experiments were designed for research purposes and were limited to laboratory settings. Further works are required to evaluate the performance of the algorithms with real scenes. Though the proposed model can only handle certain food items from unseen viewing angles at this stage, there has been a lot of progress on the basis of the existing approach (e.g., the model-based approach). To better leverage the generalization capability of deep learning, a more comprehensive 3D model database is being built to train the network. With sufficient training data, the models should then be able to handle various geometric shapes or even unseen food items that are not in the training dataset.

## 5. Conclusions

Despite current approaches showing satisfactory results in the measurement of dietary food intake from camera images, there are several limitations and constraints in the practical application of these approaches. An integrated approach based on the depth-sensing technique and deep learning view synthesis is proposed to enable accurate food volume estimation with a single depth image taken in any convenient angles. Apart from its efficiency, this technique also aims to handle food items with self-occlusion, which is one of the major challenging issues in volume estimation. We demonstrated that the proposed network architecture and point cloud completion algorithms can implicitly learn the 3D structures of various shapes and restore the occluded part of food items to allow better volume estimation. Through validating with 3D food models from the Yale–CMU–Berkeley object set, the results show that the proposed technique achieves an accuracy in volume estimation of up to 93%, which outperforms other techniques proposed in previous studies. Overall, we found that integrating different approaches could be one of the potential solutions to handle challenging issues in dietary assessment. Image-based dietary assessment will definitely play a key role in health monitoring.

## Figures and Tables

**Figure 1 nutrients-10-02005-f001:**
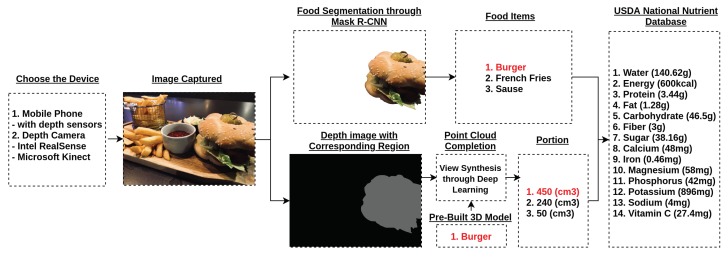
The procedure of using deep learning view synthesis to access dietary intake is shown in the figure. The burger in the colour image is recognized through Mask R-CNN. The corresponding region of the depth image is segmented out to perform deep learning view synthesis which reconstructs the point cloud of the burger. Note that the 3D model of the food items (e.g., burger) have to be pre-built prior to the view synthesis.

**Figure 2 nutrients-10-02005-f002:**
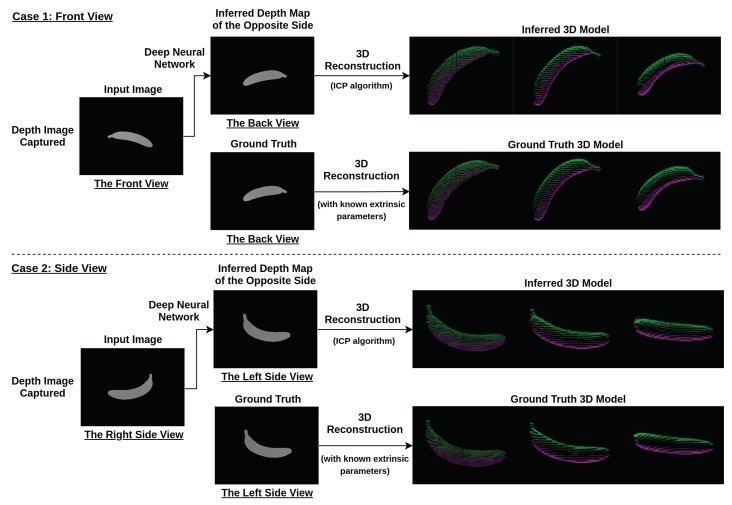
The flow chart of the process of generating a 3D point cloud of object items using the proposed algorithms. A depth image is taken from an arbitrary viewing angle (for example, from the front view (**top**) or side view (**bottom**). The image is then fed to a deep neural network to generate a depth image of the opposite view. The two depth images are then used to generate the 3D object model using the Iterative Closest Point (ICP) algorithm.

**Figure 3 nutrients-10-02005-f003:**
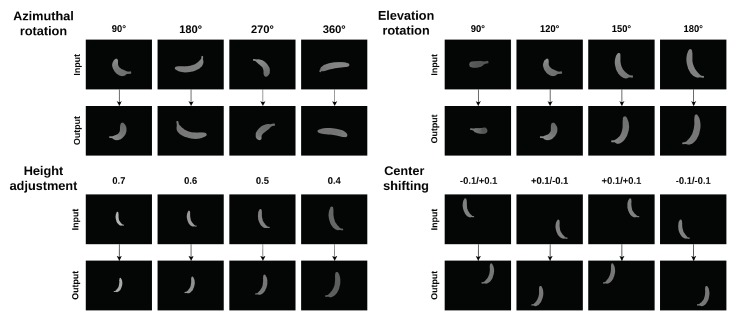
Initial (upper row) and its corresponding opposite (bottom row) depth image are shown in this figure. The images were generated through image rendering based on four types of movement. They can be listed as: azimuthal rotation, elevation rotation, height adjustment, and center shifting.

**Figure 4 nutrients-10-02005-f004:**
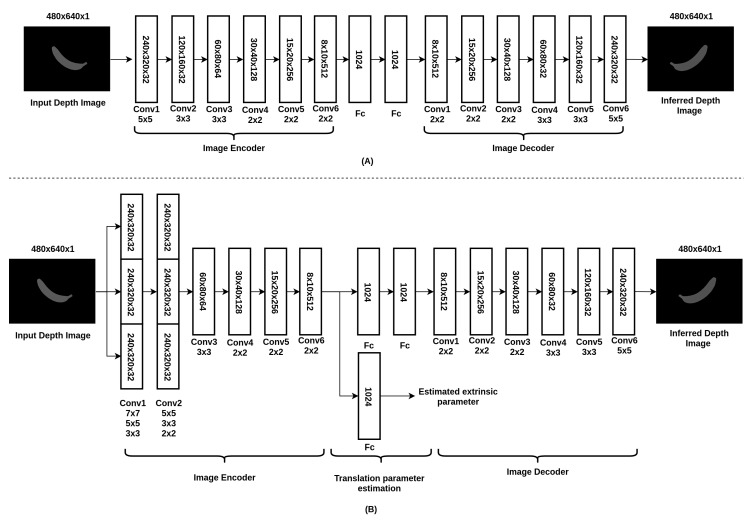
The network architecture used for depth image prediction: (**A**) The naive version of the network architecture; (**B**) the proposed network architecture with inception layers and the output for estimated extrinsic parameters.

**Figure 5 nutrients-10-02005-f005:**
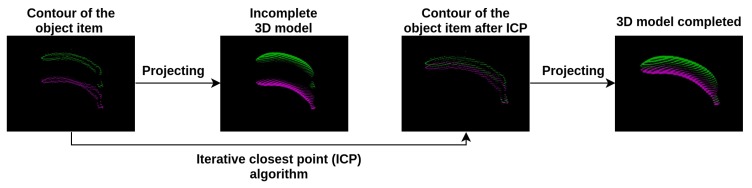
The initial and synthetic point clouds are fused together using ICP on the contours of the object items.

**Figure 6 nutrients-10-02005-f006:**
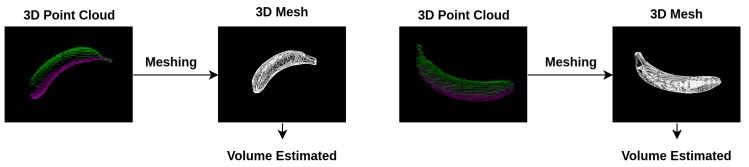
The global point cloud of a 3D model has been converted into a 3D mesh using the alpha shape.

**Figure 7 nutrients-10-02005-f007:**
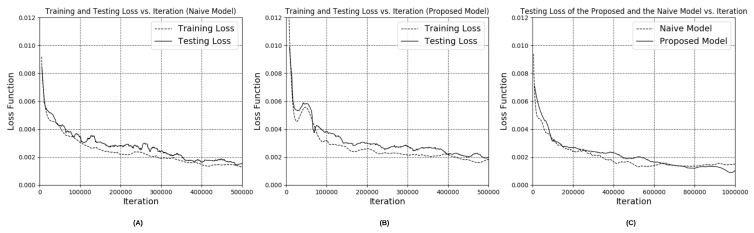
The plot of training and testing loss vs. iterations: (**A**) naive model; (**B**) proposed model; (**C**) both of the models.

**Figure 8 nutrients-10-02005-f008:**
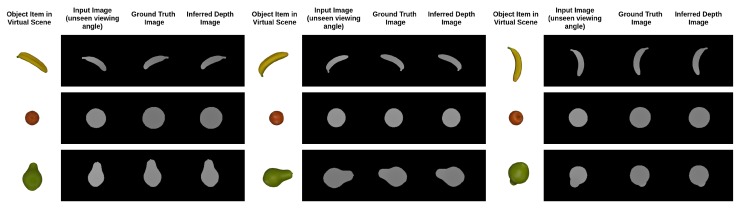
Experimental results generated by the proposed network architecture to evaluate the performance by using input images with unseen viewing angles. Each row refers to different object items. Each column refers to the input image, ground truth image, and inferred depth image respectively.

**Figure 9 nutrients-10-02005-f009:**
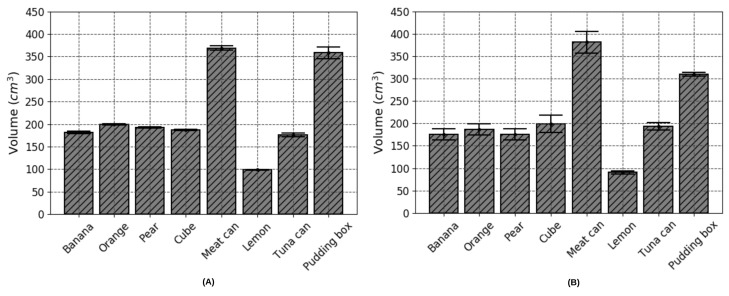
Experimental results evaluating the feasibility of the proposed algorithms: (**A**) The estimated volume for each object item based on point cloud completion using known extrinsic parameters; (**B**) the estimated volume for each object item based on ICP without using any prior knowledge.

**Figure 10 nutrients-10-02005-f010:**
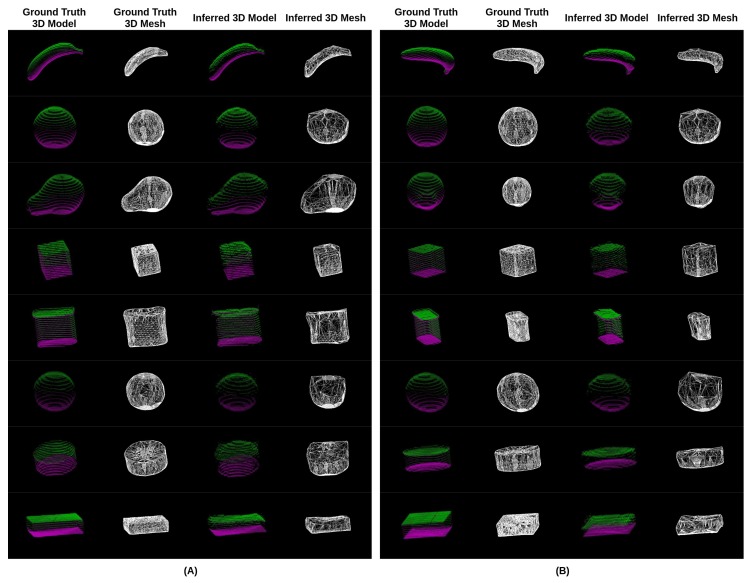
Experimental results of 3D reconstruction with objects from the Yale–CMU–Berkeley object set with unseen captured viewing angles. Depth images from two viewing angles (**A**,**B**) were used to carry out point cloud completion, respectively. (First column) The ground truth of the 3D model which was reconstructed through the proposed point cloud completion algorithm using the known extrinsic parameters. (Second column) The ground truth of the 3D mesh (Third column) The inferred 3D model based on the neural network and ICP algorithm. (Fourth column) The inferred 3D mesh.

**Table 1 nutrients-10-02005-t001:** Overall comparison among different start-of-the-art food volume estimation approaches ${}^{*}$.

Method	Author	Accuracy in Volume Estimation	Common Advantages	Common Limitations
Stereo-based approach	Gao et al. (2018) [12] Dehais et al. (2017) [9] Puri et al. (2009) [8]	80.8–88.3% 90.2–91.8% 90.5–92.0%	1. Has the ability to handle irregular food items 2. No prior knowledge required 3. No pre-built 3D models required	1. Required to capture images from different viewing angles 2. Slow processing time due to feature detection and matching 3. Unable to handle the issue of occlusion 4. Fiducial marker required
Model-based approach	Sun et al.(2015) [3] Xu et al. (2013) [4] Khanna et al. (2010) [7]	79.5% 87.7–96.4% 90.0%	1. Robust performance for certain general food shapes 2. High accuracy in pre-trained food items	1. Unable to tackle irregular food shapes 2. Pre-trained 3D model library required 3. Manual refining needed
Depth camera-based approach	Fang et al. (2016) [11]	66.1–89.0%	1. Robust performance in volume estimation 2. No fiducial marker required	1. Unable to handle the issue of occlusion 2. Depth sensing camera is not always embedded in smart devices
Deep learning approach	Christ et al. (2017) [13] Meyers et al. (2015) [10]	1.53 bread units (error in bread units) 50–400 mL (error in volume)	1. Has the ability to handle irregular food items 2. Generalization capability of neural networks	1. A large number of food images required for training 2. A high error rate in depth image prediction
Integrated approach	-	93.1%	1. Able to handle irregular food items after training 2. Able to handle occluded food items 3. No manual intervention 4. No fiducial marker required	1. A large number of food images required for training

${}^{*}$ High impact research works published between 2009 and 2018 are summarized in this table.

**Table 2 nutrients-10-02005-t002:** The range of extrinsic camera parameters used in image rendering.

	Azimuth	Elevation	Height	Shifting
Initial viewing angle	0 to 360 degree	90 to 270 degree	0.5 to 0.6 m	x: −0.1 to 0.1; y: −0.1 to 0.1
Opposite viewing angle	0 to 360 degree	270 to 450 degree	−0.5 to −0.6 m	x: −0.1 to 0.1; y: −0.1 to 0.1

**Table 3 nutrients-10-02005-t003:** Comparison of the performance of the naive version, the proposed network with extrinsic parameter prediction, and the proposed network with both inception layers and extrinsic parameter prediction.

	Naive Version	Our (Extrinsic Parameters)	Our (Inception Layers + Extrinsic Parameters)
Training Loss ${}^{a}$	0.00101	0.00093	**0.00077**
Testing Loss ${}^{b}$	0.00112	0.00104	**0.00083**
Iteration	1,000,000	1,000,000	1,000,000

${}^{a}$${}^{,}$${}^{b}$ The cost function: the lower the better.

**Table 4 nutrients-10-02005-t004:** Quantitative measurement of food items based on point cloud completion and the ICP algorithm.

Food Object Item	Ground Truth ${}^{a}$ (cm${}^{3}$)	Estimated Volume ${}^{b}$ $V_{g}$ (cm${}^{3}$)	SD of $V_{g}$ (cm${}^{3}$)	Error of $V_{g}$${}^{c}$ (%)	Estimated Volume ${}^{d}$ $V_{p}$ (cm${}^{3}$)	SD of $V_{p}$ (cm${}^{3}$)	Error of $V_{p}$ ^e^ (%)
1—Banana	-	181.6	2.8	-	175.6	12.6	**3.3**
2—Orange	203.0	199.5	1.2	1.7	186.7	12.1	**8.0**
3—Pear	-	192.75	0.9	-	175.5	12.8	**8.9**
4—Cube	185.2	187.0	0.8	1.0	198.6	19.3	**7.2**
5—Potted Meat Can	-	369.0	4.5	-	381.0	24.5	**3.3**
6—Lemon	-	98.5	1.3	-	91.0	3.4	**7.6**
7—Tuna Fish Can	180.0	176.0	4.2	2.2	193.3	8.3	**7.3**
8—Pudding Box	342.0	358.5	13.2	4.8	309.7	3.5	**9.4**

${}^{a}$ The reference volumes of the food items. ${}^{b}$ The mean estimated volume based on point cloud completion using known extrinsic parameters. ${}^{c}$ The error of the estimated volume using point cloud completion. ${}^{d}$ The mean estimated volume using the iterative closest point algorithm without prior knowledge. ^*e*^ The error of the estimated volume using the iterative closest point algorithm.

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
