# Peer review of "Food Volume Estimation Based on Deep Learning View Synthesis from a Single Depth Map"

_nutrients, 2018, doi:10.3390/nu10122005_

Round 1
Reviewer 1 Report
This paper presents an interesting and useful first step in applying 2.5D images from depth-sensing cameras now coming into common use in smart phones. The authors point out the major areas of shortcomings and the need for additional work, especially dealing with occlusion or voids. Their test samples are primarily simple near-spherical, cylindrical or box-like shapes, not the actual food servings (e.g., a pile of french fries) that would need to be addressed for this to become a practical tool. They also do not report the processing times needed to carry out the matching and calculations, which could be (probably is) an important limitation.
Author Response
Point 1: Their test samples are primarily simple near-spherical, cylindrical or box-like shapes, not the actual food servings (e.g., a pile of french fries) that would need to be addressed for this to become a practical tool.
Response 1: Thank for your comments. We agree with your comment that actual food servings should be modelled using a more complex 3D shape. The reason for carrying out those experiments can be explained in different aspects:
1. The proposed view-synthesis technique is actually built on top of current volume estimation approaches and the major purpose of the work is to resolve existing problems and limitations in food volume estimation. For instance, in model-based approach, which is considered as the state-of-the-art in food volume estimation, a pre-built 3D food model library has to be constructed before use and such model library consists of only simple shapes as well [1], [2]. Compared to model-based approach, our newly proposed technique can tackle the problem of image registration without requiring the users to rotate, translate and scale the model to match the contour of the food item in the image which undoubtedly improves the efficiency and provides better user experience.
2. The underlying reason for using simple shape in both of the model-based and our proposed technique is that complex 3D models can only be generated manually by experts. A publicly known food model library, however, does not exist nowadays yet. On the other hand, since this paper is the first study of using deep learning view synthesis to estimate food volume, most of the experiments are currently designed for research purposes and limited to laboratory settings. Once the food model library is constructed in the future, the proposed method can be extended easily to tackle various kind of food objects based on the generalization property of deep learning.
Point 2: They also do not report the processing times needed to carry out the matching and calculations, which could be (probably is) an important limitation.
Response 2: Thanks for the note. We have added the processing time into the content. Processing time is actually one of the advantages of the proposed technique. Compared to stereo-based approach, the proposed technique does not require to carry out feature matching between frames, which leads to significantly shorter processing time.
Reference
[1] Sun, Mingui, et al. "An exploratory study on a chest-worn computer for evaluation of diet, physical activity and lifestyle." Journal of healthcare engineering 6.1 (2015): 1-22.
[2] Xu, Chang, et al. "Model-based food volume estimation using 3D pose." Image Processing (ICIP), 2013 20th IEEE International Conference on. IEEE, 2013.

Reviewer 2 Report
In this paper, the authors combine the depth sensing technique with deep learning-based view synthesis for food volume estimation. It's a novel application of DL, even the network structure is noting new. The story itself is quite interesting and the paper is well written. As a reviewer, I have the following concerns,
1) How to avoid the overfitting problem? Do you employ the cross-validation strategy? Is the performance robust?
2) How to determine the regularization term λ in the cost function? Does this parameter affect the performance a lot?
3) Is it possible to compare the proposed method with the other volume estimation methods, even they do not employ the depth information? The readers may want to know how much is the improvement.
4) I suggest you express 'multiply' as '*' or 'x' (not italic).
Author Response
Point 1: How to avoid the over-fitting problem? Do you employ the cross-validation strategy? Is the performance robust?
Response 1: Thanks for your questions. Yes. To examine the feasibility of the proposed network in inferring depth image from unseen viewing angles and avoid over-fitting, it is essential to carry out a thorough evaluation of the proposed technique. Similar to the method used in prior works of point cloud completion [1], [2], [3], holdout method, a simple kind of cross validation, is used to evaluate the performance of the model in which 70% (20k images per object item) of the rendered depth images are used to train the neural network, while 10\% (2.85k images per object item) and 20% (5.71k images per object item) of the images with unseen viewing angles are selected as the validation dataset and testing dataset respectively (Two well-known deep learning network architectures, ResNet [4] and VGGnet [5], have been evaluated using this method as well). The detailed information about image splitting has been added to the paper to explain the rationale. Also, the graph of training and testing loss versus iteration is added to the paper. As shown in Figure 7, the testing loss is comparable to the training loss which represents that the model is able to handle object items with unseen viewing angles and is not suffering from over-fitting problem.
For the performance evaluation, it has shown that our proposed network and the ICP algorithm are robust in handling depth image with unseen viewing angles and point cloud completion respectively (The error bar has been presented in Figure 9 to show the robustness).
Point 2: How to determine the regularization term λ in the cost function? Does this parameter affect the performance a lot?
Response 2: Thanks for the note. λ is the regularization term added to the cost function to better train the neural network. In this work, λ is determined empirically. The selection of λ will affect the convergence speed of the network (λ=10-3 is used in the cost function).
Point 3: Is it possible to compare the proposed method with the other volume estimation methods, even they do not employ the depth information? The readers may want to know how much is the improvement.
Response 3: Thanks for your suggestion. We added the detailed information of other proposed methods and compared their performance with our method in terms of the accuracy of volume estimation in the table.
Point 4: I suggest you express 'multiply' as '*' or 'x' (not italic).
Response 4: Thanks for your suggestion. We have made the revision accordingly.
Reference
[1] Tatarchenko, Maxim, Alexey Dosovitskiy, and Thomas Brox. "Multi-view 3d models from single images with a convolutional network." European Conference on Computer Vision. Springer, Cham, 2016.
[2] Varley, Jacob, et al. "Shape completion enabled robotic grasping." Intelligent Robots and Systems (IROS), 2017 IEEE/RSJ International Conference on. IEEE, 2017.
[3] Zelek, John, and Nolan Lunscher. "Point cloud completion of foot shape from a single depth map for fit matching using deep learning view synthesis." Computer Vision Workshop (ICCVW), 2017 IEEE International Conference on. IEEE, 2017.
[4] He, Kaiming, et al. "Deep residual learning for image recognition." Proceedings of the IEEE conference on computer vision and pattern recognition. 2016.
[5] Simonyan, Karen, and Andrew Zisserman. "Very deep convolutional networks for large-scale image recognition." arXiv preprint arXiv:1409.1556 (2014).

Reviewer 3 Report
The authors proposed a view synthesis based approach for 3D point cloud reconstruction of rigid objects with examples of food items. Food volume estimation is a challenging problem and is an important step towards accurate dietary assessment. The authors provided a good review of existing techniques for food volume estimation. However, the technical contribution is limited and there are some concerns regarding the proposed work:
- The paper uses an existing object dataset with 3D models. Only 8 objects from this dataset are reported, perhaps these are the only food related objects. These food objects have simple shapes of sphere, cylinder and box. It is not clear how the proposed method can be used for everyday consumed foods which often have deformable shapes that will not work with simple 3D models.
- There are some mistakes in Equation (1), the ~ should be placed on d, not u. Also, the cost function should sum over all depth images.
- What is the value of lamda used in Equation (1)?
- On line 196, there is a typo, v(d,v) should be d(u,v).
- Table 3 shows the improvement of training and testing loss for the proposed method. It is not sufficient to show an instance of the loss, but rather the author should show a plot of the loss for different methods vs epochs which is commonly done in the literature to show performance of neural networks.
- The depth images in the paper are of high quality and free of background noise. It is not clear how the proposed method can handle depth image from consumer devices.
- It is not clear how the proposed method can be used in dietary assessment studies as it would be impossible to reconstruct 3D models for different food objects that are not in the dataset used in this paper.
Author Response
Point 1: The paper uses an existing object dataset with 3D models. Only 8 objects from this dataset are reported, perhaps these are the only food related objects. These food objects have simple shapes of sphere, cylinder and box. It is not clear how the proposed method can be used for everyday consumed foods which often have deformable shapes that will not work with simple 3D models.
Response 1: Thank for your comments. We agree with your comment that actual food servings should be modelled using more complex 3D shapes. The reason for carrying out those experiments can be explained in different aspects:
1. The proposed view-synthesis technique is built on top of current volume estimation approaches and the main purpose of the work is to address the existing problems and limitations of food volume estimation. For instance, in model-based approach, which is considered as the state-of-the-art in food volume estimation, a pre-built 3D food model library has to be constructed beforehand, and such model library consists only simple shapes as well [1]. Compared to model-based approach, our newly proposed technique can tackle the problem of image registration without requiring the users to rotate, translate and scale to match the contour of the food item on the image which undoubtedly improves the efficiency and provides better user experience.
2. The underlying reason for using simple shape in both of the model-based and our proposed technique is that complex 3D models can only be generated manually by experts. A publicly available food model library, however, does not exist yet. On the other hand, since this paper is the first study of using deep learning view synthesis to estimate food volume, most of the experiments are currently designed for research purposes and limited to laboratory settings. If an extensive food model library is constructed in the future, the proposed can be easily extended to tackle various kind of food objects based on the generalization ability of deep learning.
To explain how the proposed method can be used to achieve dietary assessment, more detailed information and figures are added to the paper.
Point 2: There are some mistakes in Equation (1), the ~ should be placed on d, not u. Also, the cost function should sum over all depth images. What is the value of lamda used in Equation (1)? On line 196, there is a typo, v(d,v) should be d(u,v).
Response 2: Thanks for the reminder. The typo and mistake have been revised accordingly. λ is the regularization term added to the cost function to better train the neural network. In this work, λ is determined empirically. λ=10-3 is used in the cost function.
Point 3: Table 3 shows the improvement of training and testing loss for the proposed method. It is not sufficient to show an instance of the loss, but rather the author should show a plot of the loss for different methods vs epochs which is commonly done in the literature to show performance of neural networks.
Response 3: Thanks for your suggestion. The graphs of training and testing loss vs. iteration are added to the paper. It has shown in the figures that the testing loss is comparable to the training loss which indicates that the models are capable of handling object items with unseen viewing angles and without any over-fitting problem. Furthermore, a plot of the loss for different methods is added as well to compare the performance of the naive model and the model with inception layers and extrinsic parameters.
Point 4: The depth images in the paper are of high quality and free of background noise. It is not clear how the proposed method can handle depth image from consumer devices.
Response 4: Thanks for your comments. This work mainly focuses on the application of deep learning view synthesis in volume estimation, so that depth image filtering is not discussed in detail in this paper. Normally, bilateral filter is used to remove outliers in the depth image and statistical outlier removal filter is designed for removing noisy data points in 3D point clouds. To better show how the previous filtering method can be used with the proposed technique, descriptions and references are added to the content of the paper to explain the details.
Point 5: It is not clear how the proposed method can be used in dietary assessment studies as it would be impossible to reconstruct 3D models for different food objects that are not in the dataset used in this paper.
Response 5: Thanks for your comments. More detailed information have been added to the content for better illustration. In addition, since this is the first study of using deep learning view synthesis to estimate food volume, most of the experiments are currently designed for the investigative research purposes and which is limited to laboratory settings. Further works will be required to evaluate the performance of the algorithms with real scenes. Though the proposed model can only handle a limited number of food items from unseen viewing angles at this stage, it could be a major step forward from the existing approach (e.g. model-based approach) for food volume estimation. Aforementioned, a more comprehensive 3D model database could be built to train the network in order to better leverage the generalization capability of deep learning. With sufficient training data, the models should then be able to handle various geometric shapes or even unseen food items which are not in the training dataset.
Reference
[1] Sun, Mingui, et al. "An exploratory study on a chest-worn computer for evaluation of diet, physical activity and lifestyle." Journal of healthcare engineering 6.1 (2015): 1-22.
[2] Xu, Chang, et al. "Model-based food volume estimation using 3D pose." Image Processing (ICIP), 2013 20th IEEE International Conference on. IEEE, 2013.

Round 2
Reviewer 2 Report
The authors have addressed all my concerns.
Author Response
Thank you for your review and comments.
Reviewer 3 Report
The authors have addressed all comments from the previous review.
However, in the newly added Figure 1, it is not clear how the proposed view synthesis method can handle objects such as the burger in the example image. Will the authors create a new dataset with 3D models of foods? The continuation of this work will add value to food volume estimation problem that is an essential part of image-based dietary assessment methods.
Author Response
Response to Reviewer 3 Comments
Point 1: However, in the newly added Figure 1, it is not clear how the proposed view synthesis method can handle objects such as the burger in the example image.
Response 1: Thanks for your comments. More detailed information has been added to Figure 1 for better illustration (e.g. the pre-built 3D models). In the figure, it shows that 3D model of the food items (e.g. burger) have to be pre-built prior to the view synthesis.
Point 2: Will the authors create a new dataset with 3D models of foods?
Response 2: Thanks for your question. Yes. Our group is planning to create a more comprehensive 3D dataset of food items. We believe our model should then be able to handle a large variety of food items in the coming future.